# Cells grown in three-dimensional spheroids mirror in vivo metabolic response of epithelial cells

Simon Lagies [1,2,3], Manuel Schlimpert[1,2,3,6], Simon Neumann[4], Astrid Wäldin[4], Bernd Kammerer[1,2,5], Christoph Borner[2,4,5] & Lukas Peintner [4✉]

Metabolism in cells adapts quickly to changes in nutrient availability and cellular differentiation status, including growth conditions in cell culture settings. The last decade saw a vast increase in three-dimensional (3D) cell culture techniques, engendering spheroids and organoids. These methods were established to improve comparability to in vivo situations, differentiation processes and growth modalities. How far spheroids mimic in vivo metabolism, however, remains enigmatic. Here, to our knowledge, we compare for the first time metabolic fingerprints between cells grown as a single layer or as spheroids with freshly isolated in situ tissue. While conventionally grown cells express elevated levels of glycolysis intermediates, amino acids and lipids, these levels were significantly lower in spheroids and freshly isolated primary tissues. Furthermore, spheroids differentiate and start to produce metabolites typical for their tissue of origin. 3D grown cells bear many metabolic similarities to the original tissue, recommending animal testing to be replaced by 3D culture techniques.

[1] Center for Biological Systems Analysis (ZBSA), Albert-Ludwigs-University Freiburg, Habsburgerstrasse 49, 79104 Freiburg, Germany. [2] Spemann Graduate School of Biology and Medicine, Albert-Ludwigs-University Freiburg, Albertstrasse 19a, 79104 Freiburg, Germany. [3] Faculty of Biology, Albert-Ludwigs-University Freiburg, Freiburg, Germany. [4] Institute of Molecular Medicine and Cell Research, Albert-Ludwigs-University Freiburg, Stefan Meier Strasse 17, 79104 Freiburg, Germany. [5] BIOSS Centre of Biological Signalling Studies, Albert-Ludwigs-University Freiburg, Schaenzlestrasse 18, 79104 Freiburg, Germany. [6] Present address: Agilent Technologies Deutschland, Hewlett-Packard-Strasse 8, 76337 Waldbronn, Germany. ✉email: lukas.peintner@mol-med.uni-freiburg.de

Living cells operate an intricate network of anabolic and catabolic processes that provide the optimal nutrient and building block balance for each step in a cell's life. All these pathways produce specific intermediates that can adapt rapidly to changes in the physiological conditions of a cell. Mapping these intermediates, that are metabolites, produces the metabolic fingerprint of a cell that further may be compared to other cells in different environments. Hence, monitoring the metabolome of a cell is a valid approach to monitor the general wellbeing of a cell without the need to focus on the activity of individual molecular pathways[1].

Cultured cells actively undergoing cell proliferation have a high demand in substrates to maintain their growth in biomass. Logically, the central regulators of the cell cycle such as the CDK/pRB/E2F system, have a direct influence on the cell's metabolism[2]. For instance, E2F is a transcription factor for the 6-phosphofructo-2-kinase/fructose-2,6-bisphosphatase (PFKFB) isoenzyme, that plays a crucial role in the activation of glycolysis. Furthermore, E2F4 is important for mitochondria biogenesis and blocks cytochrome c and ferredoxin reductase to diminish oxidative phosphorylation during cell cycle progression[3,4]. E2F1 also blocks the function of pyruvate dehydrogenase kinase that controls pyruvate dehydrogenase activity and thus enhances glucose oxidation[3,5].

E3 ligases such as the anaphase promoting complex/cyclosome (APC/C) and the Skp1/Cullin/F-box (SCF) have been shown to be involved in the degradation of key enzymes in glycolysis (PFKFB3) and glutaminolysis (glutaminase 1)[6]. When APC/C disappears at the end of G1, glycolysis and glutaminolysis face a rush and provide the needed carbohydrates and nitrogen-containing compounds for a successful transition into S-phase.

Quiescent cells are in a dormant, non-proliferative state[7] that can be induced by contact inhibition, lack of nutrients, telomere shortening, or other factors[8]. The metabolome undergoes drastic fluctuations when cells leave the cell cycle, since the demand for building blocks and metabolites changes to adapt to the new situation. While the metabolism in starving cells is well understood[7], metabolic changes after contact inhibition are only ill-defined. Upon contact inhibition, the cyclin-dependent kinase inhibitor $p27^{Kip1}$ is upregulated[9]. This causes a change in the metabolic flux, and citrate is not metabolized in the tricarboxylic acid cycle to α-ketoglutarate but rather shuttled into the cytosol where it is involved in the formation of fatty acids or histone modifications[10]. Furthermore, the rate of glucose uptake and lactate secretion is decreased by 50% in fibroblasts after contact inhibition as it happens in three-dimensional cell culture. Once cells form spheroids, nutrient availability changes across the diameter of the sphere. While the outer layers of cells get the full nutrient coverage from the medium, inner cells face drastic shortages in nutrients, depending on the size of the spheroid[11,12]. When investigating nutrient availability, the PI3K-AKT-mTOR pathway attracted the utmost attention in the last decade as a mean to activate autophagy[13]. While it does not play a role in metabolic shifts induced by contact inhibition, it is of central importance once nutrients are unevenly distributed among the cell mass. Starving cells modulate mTOR signaling via an activated Her2 and therefore have high autophagic activity that enables the recycling of building blocks to maintain cellular functions[11,14].

To learn about the basic principles of life, researchers exploit cell culture systems. Based on findings yielded in these artificial systems, new compounds and drugs are generated to treat diseases. However, to fully understand the effect of a new drug, animal models are the tool of choice to screen for possible side effects before administering them to humans[15]. Over the last decades, there have been increasing ethical issues concerning this practice, which resulted in the "Replace–Reduce–Refine" recommendations, now endorsed by most regulatory authorities[16].

In order to replace animal testing, researchers have developed multiple new tools and methods[17–19]. Next to simulating organ functions on multi-chambered cassettes, so-called organ-on-a-chip technologies, the main focus was improvement of standard cell culture that finally led to cell culture practices growing in three dimensions[20,21]. Three-dimensional cultures have vastly grown in popularity for the last 20 years, mostly because of their presumed similarity to the in vivo situations but also because of the easy to handle concept[22]. However, solid proof that cells growing in 3D cell culture behave similar to tissues or organs is still very much lacking. Here we show for the—to our knowledge—first time the direct comparison of established renal cell lines growing in conventional 2D cell culture or in 3D spheroids to cells freshly isolated from murine kidney tissue. We find that 3D grown cells have striking similarities to cells isolated from the distal tubule but also some differences based on immortalization or adaptation to long time cell culture condition.

## Results

**mIMCD3 cells grow in 3D spheroids and reduce proliferation.** Immortalized mouse inner medullary collecting duct (mIMCD3) cells are typically kept in standard 2D cell culture conditions and show exponential growth until they reach confluency[23]. Upon establishing dense cell–cell contacts, mIMCD3 cells stop proliferating and differentiate, ultimately forming a primary cilium originating from the centrosome that moves from the nucleus area to the apical cell surface[24]. In the absence of a solid surface for the cell to attach to the matrix, adherent cells die by anoikis[25]. This form of cell death is abrogated when cells get in close contact to each other. In 3D culture, stable cell–cell contacts form spontaneously, cells clump together, form spheroids, and anoikis-induced cell death signal is blocked (Fig. 1a). Upon blockage of cell–matrix and cell–cell contacts by drugs such as the fungal secondary metabolite gliotoxin, cells cannot form spheroids and are doomed to die although they are still in physical contact to each other (Supplementary Fig. 1a)[25]. A method to induce a surface-free environment for cells allowing for cell–cell contact is provided by the hanging drop approach[26–28]. Cells in a suspension are put dropwise on an inverted plate. The gravity pulls the floating cells to the lowest spot of the drop, where they get into physical contact to each other and form spheroids within several minutes[26,27]. Cells in such spheroids stop proliferating and start differentiating as evidenced by an accumulation of cells in the G1 phase of the cell cycle (Fig. 1b and Supplementary Fig. 1b) and a diminished Ki67 signal, a bona fide marker for cell proliferation (Fig. 1c).

While performing these experiments, we noted that the same number of cells growing in 2D induced a change of the medium color from red to yellow much faster than cells growing in 3D. Such a medium color change is due to acidification caused by small organic acids. A prominent, but not sole organic acid excreted by the cell is lactic acid. This acidification prompted us to monitor the changes of the metabolome in cells grown in either standard 2D cell culture or 3D spheroids.

**mIMCD3 spheroids display a distinct metabolic pattern.** Changes in cell cycle and differentiation status are often accompanied by massive changes in the metabolism of the cells[29]. To measure the metabolic fingerprint of mIMCD3 cells grown either in 2D or as 3D spheroids, cell lysates were analyzed by untargeted gas chromatography/mass spectrometry (GC/MS)-based metabolic screening. Principal component analysis (PCA) revealed

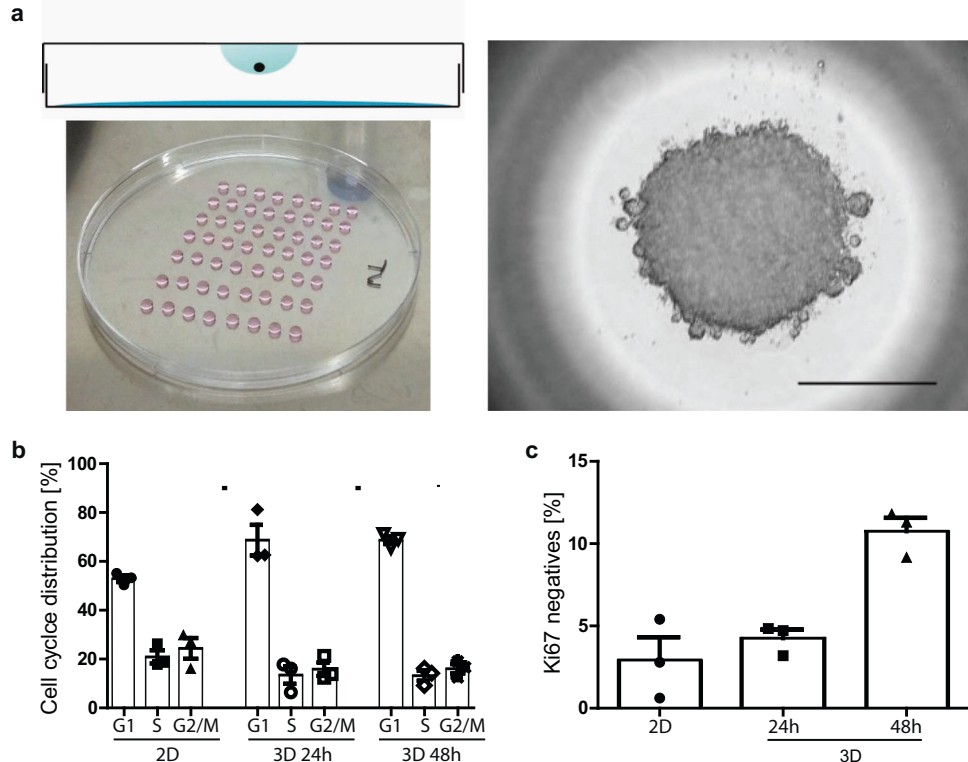

**Fig. 1 mIMCD3 cells change their cell cycle ratio when grown in 3D culture. a** Cells growing in a contact-free environment (e.g., a hanging drop from the lid of a cell culture vessel) spontaneously form spheroids. Size marker = 500 μm. **b** 2D grown cells and cells in 3D culture for 24 and 48 h change their cell cycle distribution over time. Cells in G1 phase expand on the expense of cells in S phase and G2/M phase. **c** Ki67 serves as a bona fide marker for cells actively participating in cell cycle. Cells negative for Ki67 represent G0 phase cells entered the dormant state. Bars represent mean ± standard deviation of three biologically independent FACS experiments.

that principal component (PC) 1 discriminated between 2D (negative PC1) and 3D grown cells (positive PC1) with 53.9% of the observed differences (Fig. 2a).

A heat map of metabolites with a false discovery rate (FDR)-corrected $q$-value < 0.05 (statistical results are shown in Supplementary Data 6) shows the main differences of metabolites in a Ward clustering (Fig. 2b). These differences are in line with the global discrimination of 2D and 3D grown cells by PC1 (Fig. 2a). In total, nine metabolites were upregulated and 15 metabolites were downregulated in cells grown in 3D. The contributing metabolites were products or intermediates of anabolic and catabolic pathways like the tricarboxylic acid cycle and glycolysis as well as amino acid and fatty acid metabolism. Intermediates of glycolysis were concomitantly decreased in 3D grown cells. The essential amino acids phenylalanine, threonine, and valine were shown to be downregulated, while leucine seemed to be upregulated in cells grown in 3D (Fig. 2b). Furthermore, building blocks of the lipid metabolism such as diethanolamine, palmitic acid, oleic acid, or ethanolamine-phosphate were differentially regulated after changing growth conditions (Fig. 2b). These alterations suggest differentially regulated lipid metabolism between the two growth conditions.

Following our initial observation of faster acidification, we analyzed conditioned media from conventionally grown cells and those grown in hanging droplets. Confirmatory to the endometabolic results, global differences were revealed in the cell culture supernatant (Fig. 2c). Importantly, lactic acid was significantly increased in medium obtained from 2D cultures ($q$-value: 0.0002), which is in line with the increased glycolysis in two-dimensionally grown mIMCD3 cells and further explains the faster medium color change (Fig. 2d, corresponding statistical results are displayed in

Supplementary Data 7). A further decrease of extracellular glucose was not observed ($q$-value glucose: 0.6409).

**Spheroids are metabolically similar to nephron epithelium**. The previous experiment showed that the metabolism of individual cell lines is highly versatile and rapidly adapts to changes in environment and growth conditions. mIMCD3 cells are kidney epithelial cells. Thus, we were very curious if these cells grown in 3D show a similar metabolic pattern as cells freshly isolated from murine nephrons or whole kidneys. Based on the PCA we could discriminate between mIMCD3 cells grown in 2D or 3D, freshly isolated nephrons, and whole kidneys. Interestingly, PC1 and PC2 revealed that nephrons are metabolically more similar to mIMCD3 cells grown in 3D than to those grown in 2D (Fig. 3a).

A heat map of all significantly changed metabolites (statistical results are shown in Supplementary Data 8) confirmed the global differences in metabolites observed in Fig. 2. The most obvious difference was detected in the metabolic profile of cells grown in 2D in comparison to the other three conditions. These cells presented a strong upregulation of many metabolites correlated with cell growth such as glycolysis intermediates, oxidative phosphorylation, spermidine, ATP degradation products, lipid metabolism, and various amino acids. This pattern was very similar to our previous measurement shown in Fig. 2b. Additionally, a biochemical in-depth analysis with cell lysates grown in the respective conditions confirmed the change in glycolysis. The levels of hexokinase 2 were diminished on the protein (Fig. 4a and Supplementary Fig. 4) and the mRNA (Fig. 4c) level in 3D spheroids and nephron and kidney cells. Also the amount of glucose-6-phosphate dehydrogenase (G6PD) was decreased in 3D grown cells and cells freshly isolated

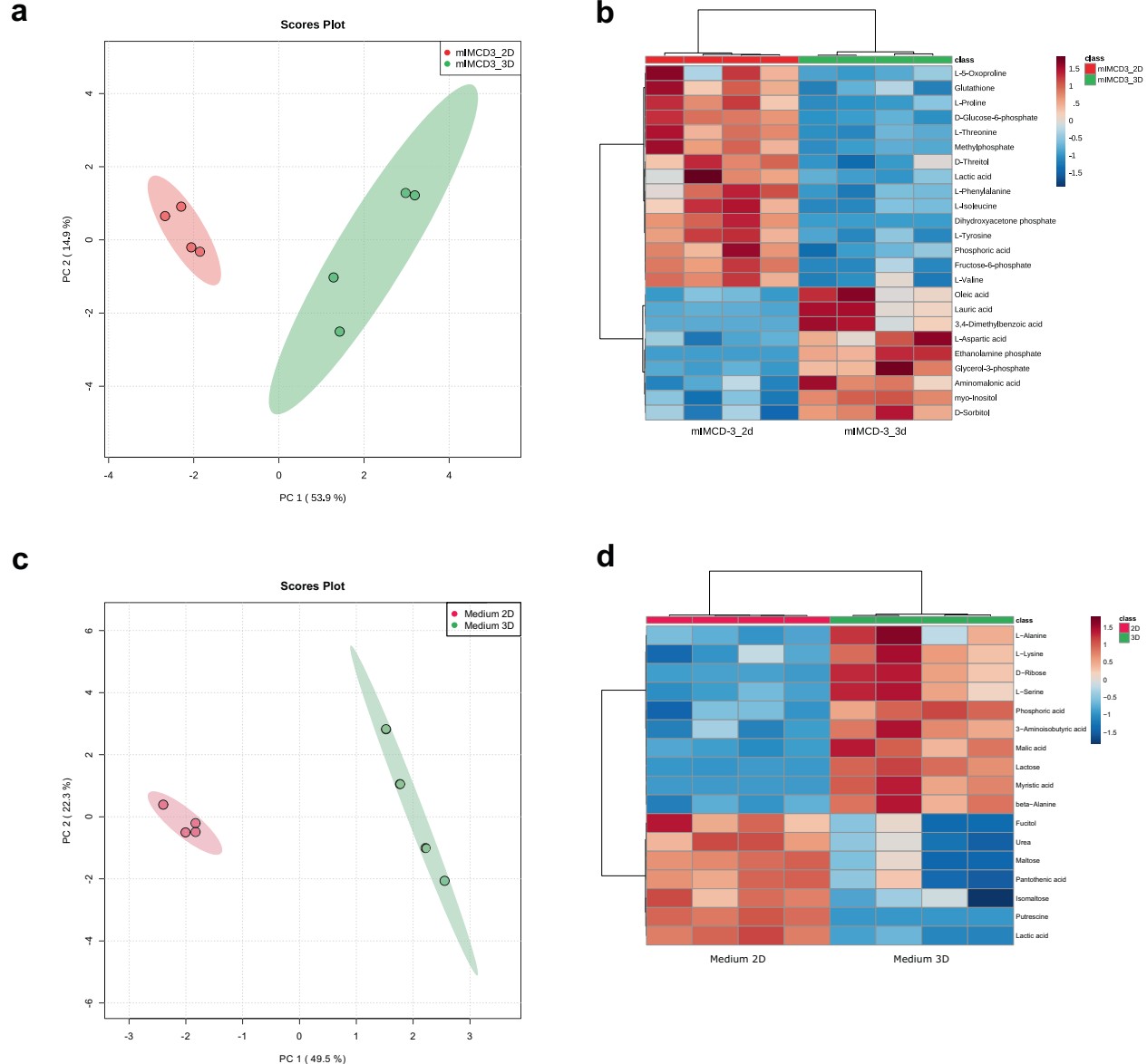

**Fig. 2 The endometabolome of mIMCD3 cells growing in 3D differs to cells growing in 2D. a** Principal component analysis (PCA) of endometabolite profiling with PC1 against PC2. Red: cells grown in 2D, green: cells grown in 3D. Replicates are highlighted as individual dots; shaded area shows the 95% confidence interval. **b** Heat map and cluster analysis of endometabolites reveal differences between cells grown in 2D or 3D. Range-scaled z-scores of annotated features with a q-value < 0.05. Differently abundant metabolite clusters after Pearson and Ward reflect the similarity between cultured 2D or 3D grown mIMCD3 cells. Red: 2D, green: 3D grown cells, each n = 4. **c** PCA of exometabolome analysis with PC1 against PC2. Red: cells grown in 2D, green: cells grown in 3D. Replicates are highlighted as individual dots; shaded area shows the 95% confidence interval. **d** Heat map and cluster analysis of exometabolites reveal differences between cell-conditioned media obtained from cells grown in 2D or 3D. Range-scaled z-scores of annotated features with a q-value < 0.05. Differently abundant metabolite clusters after Pearson and Ward reflect the similarity between cultured 2D or 3D grown mIMCD3 cells. Red: 2D, green: 3D grown cells, each n = 4.

from the kidney (Fig. 4d). G6PD is a crucial enzyme of the pentose phosphate pathway fueling nucleotide synthesis. Its decrease in 3D grown cells is in accordance to the diminished Ki67 signal and a faithful reporter for the exit of cells from active cell cycle.

Furthermore, we identified important renal metabolites up-regulated in 3D grown cells and freshly isolated nephrons. Betaine, for instance, is a metabolite strongly accumulated in cells from the distal tubule[30]. We saw an accumulation of betaine in 3D grown mIMCD3 cells, similar to the level of isolated nephrons (Fig. 3b). In the total kidney lysate, however, this elevated level of betaine could not be recorded, since the fraction of distal tubules in the kidney is too small to contribute a significant share. In the

whole kidney lysate, taurine serves as the predominant marker for the kidney signature. To confirm our GC/MS analysis, we determined the expression levels of bgt-1 (slc6a12, betaine/GABA transporter) that mediates the accumulation of betaine in the cell. These levels were significantly higher (p < 0.0001) in cells grown in 2D as compared to the other three conditions on both the mRNA (Fig. 4b) and on the protein level (Fig. 4a). As the cell culture medium does not contain any additional betaine, we speculate that synthesis is the major contributor of betaine levels in 3D grown cells. Additionally, the cell cycle distribution of mIMCD3 cells was similar to nephrons (Supplementary Fig. 1c). Inconclusive, however, was the regulation of ethanolamine in the

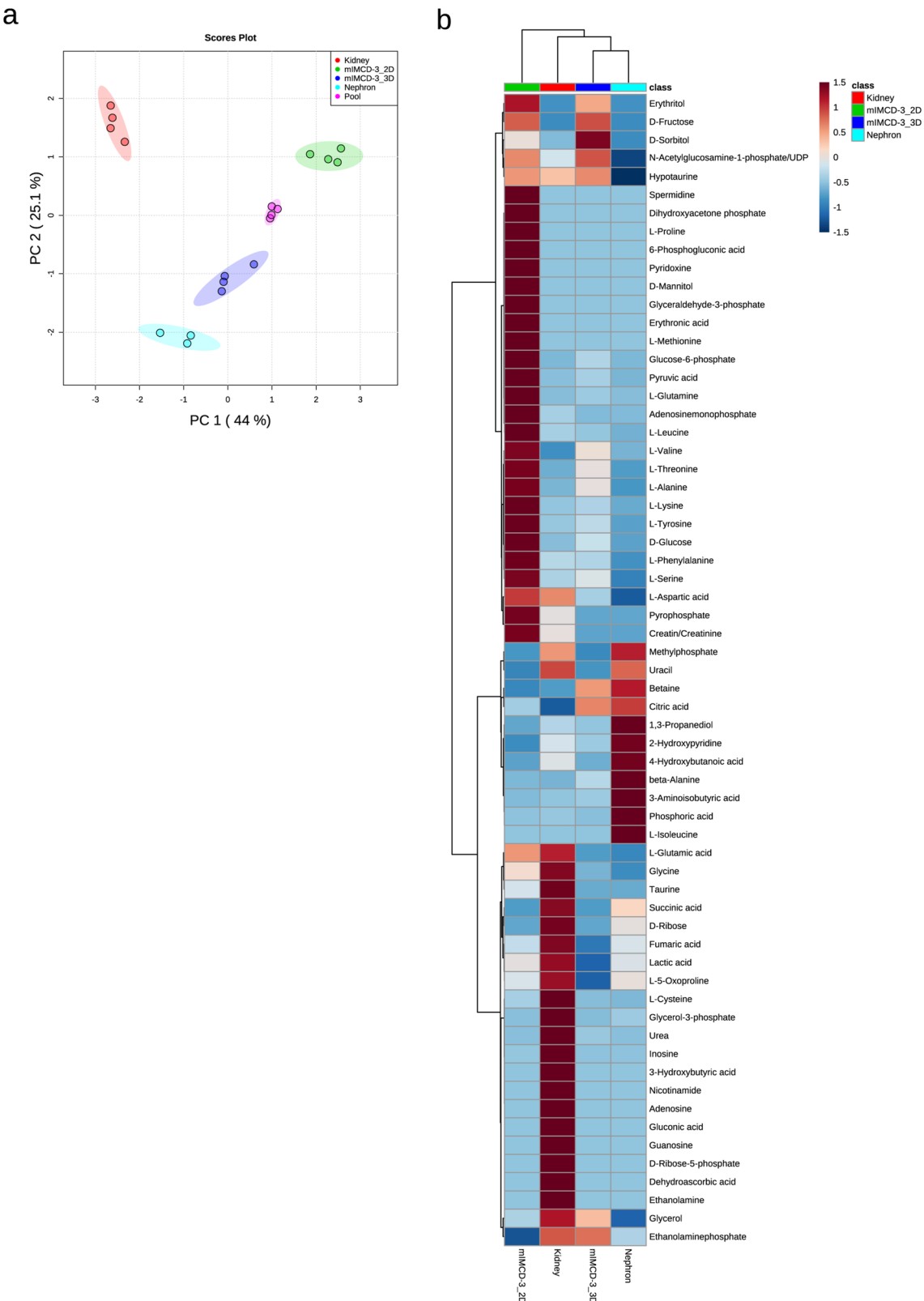

**Fig. 3 The endometabolome of mIMCD3 cells and freshly isolated kidney cells. a** PCA of endometabolite profiling with PC1 against PC2. Red: whole kidney lysate, green: cells grown in 2D, blue: cells grown in 3D, turquoise: isolated distal tubules from perfused kidneys, and pink: pooled quality control sample. Replicates are highlighted as individual dots; shaded area shows the 95% confidence interval. **b** Heat map and cluster analysis of endometabolites. Range-scaled $z$-scores of annotated features with a $q$-value < 0.05 according to ANOVA and FDR correction. Differently abundant metabolite clusters after Pearson and Ward reflect the differences between cultured 2D or 3D grown mIMCD3 cells and kidney lysates. Green: 2D grown cells, red: whole kidney lysate, blue: 3D grown mIMCD3 cells, turquoise: isolated nephrons, $n$ for nephrons = 3, $n$ for 2D, 3D, and kidneys = 4.

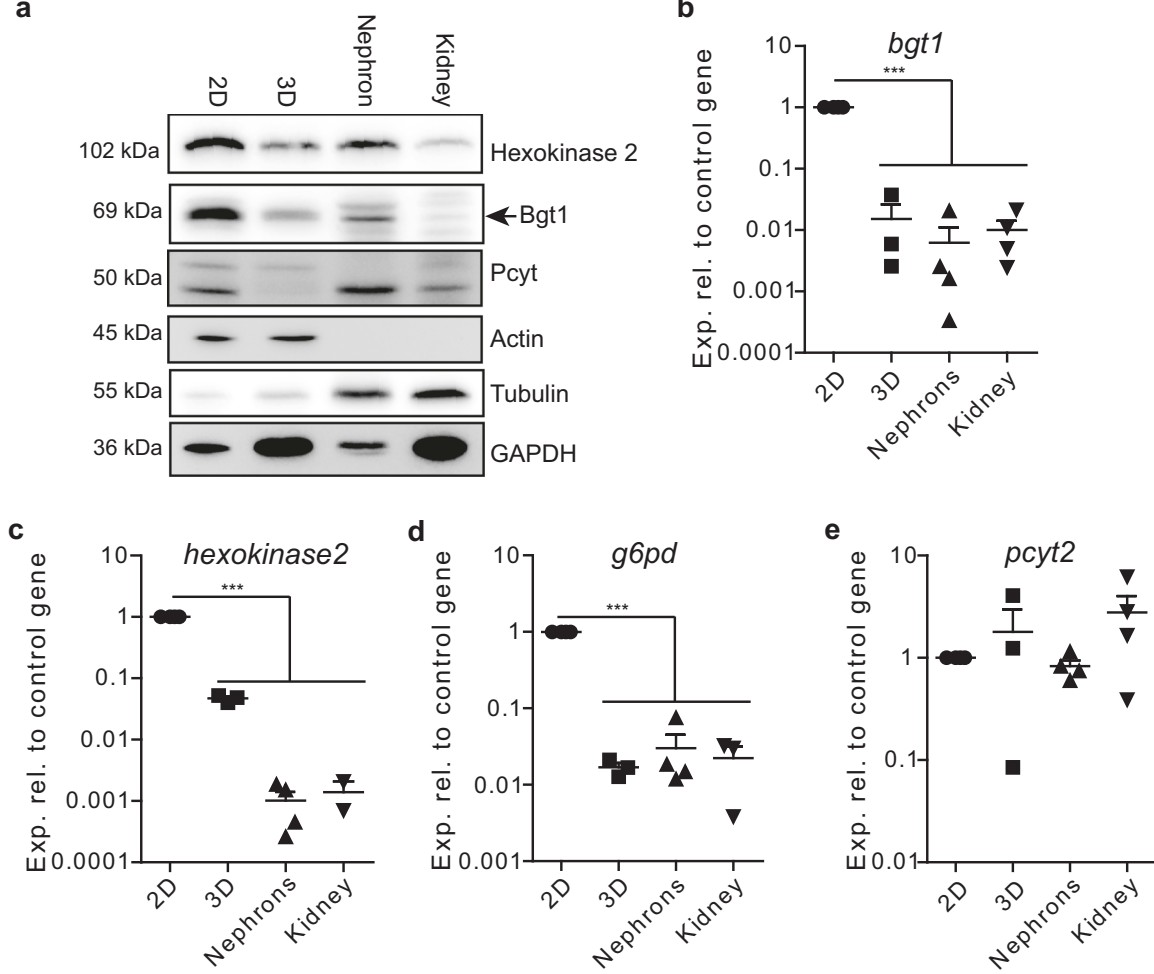

**Fig. 4 The endometabolome is shaped by the activity of enzymes. a** Western blot analysis of cells grown in 2D or 3D and of lysates isolated from whole kidney or isolated nephrons on key enzymes in the metabolism such as hexokinase-2, bgt-1 (arrowhead), and pcyt2α. Actin and tubulin served as loading controls, since GAPDH, as a member of the glycolysis pathway was not reliable as housekeeping protein. **b–e** RNA expression of selected enzymes was analyzed for bgt-1 (**b**), hexokinase-2 (**c**), glucose-6-phosphate dehydrogenase (**d**), and pcyt1α (**e**). Expression of actin served as a control gene, bars represent mean ± standard deviation, $n$ for 3D = 3, n for 2D, nephrons, and kidneys = 4. ***$p < 0.0001$ of 2D values against all three other situations.

kidney. While the GC/MS analysis reported a strong upregulation of ethanolamine, the regulating enzyme PCYT2 was not correspondingly regulated on the mRNA and protein level (Fig. 4a, e).

In our GC/MS analysis, we used methanol/water extractions to isolate metabolites. However, this method limits the detection and characterization of lipids and fatty acids. Nevertheless, some hits in Figs. 2b and 3b assume changes in lipid composition after change in growth conditions. To further investigate this issue, we re-extracted the cell pellets with chloroform to isolate the lipids content. A PCA revealed that in PC1 mIMCD3 cells grown either in 2D or 3D settle on the positive side of the axis, while the lysates from nephrons and kidneys move to the negative side of the graph. Importantly, PC2 shows a clustering of cells isolated from nephrons and mIMCD3 grown in 3D conditions (Fig. 5a).

An in-depth analysis and clustering of all the identified and significantly altered lipids (see Supplementary Data 9) allows the pooling of various lipids into six clusters (a–f, Fig. 5b). A detailed description of the different clusters can be found in Supplementary Fig. 3. Cluster a represents largely phosphatidylcholines (PC), phosphatidylethanolamines (PE), and phosphatidylserines with large fatty acids and quite an amount of polyunsaturated fatty acids. Cluster b consists mainly of cardiolipins, cluster c contains diverse lipids. Cluster d involves mostly plasmalogenes and cluster

e and f contain specific PE and PCs. In clusters a and b, a clear discrimination between mIMCD3 cells and kidney isolated lipids can be seen. Isolated nephrons and 3D grown mIMCD3 cells show a strong conformance in cluster c, containing lipids such as 1-α,24R,25-trihydroxyvitamin D2, 25-hydroxyvitamin D3-26, 23-lactone or 1,25-dihydroxyvitamin D2, all intermediates of renal vitamin D metabolism.

## Discussion

One major incentive for research in 3D cell cultures is the urge to reduce animal testing. However, isolated cells in culture often react entirely different to drug exposure than cells within animals[31]. The reasons for that may be manifold but changes in nutrient and oxygen availability certainly play a major role[18]. 3D cell culture aims to better mimic the in vivo situation without the need of maintaining an animal facility. In many cases, the efficiency of this approach has been proven[32]. Here, we analyzed the metabolome of cells grown as spheroids (Fig. 2) and compared it to the metabolic fingerprint of cells isolated from mice. Freshly isolated cells from murine distal ducts revealed an analogous metabolome as cells grown in spheroids (Fig. 3a). Cells grown in standard 2D culture, however, vastly differed from the signature of isolated nephrons (Fig. 3b). This allows the conclusion that

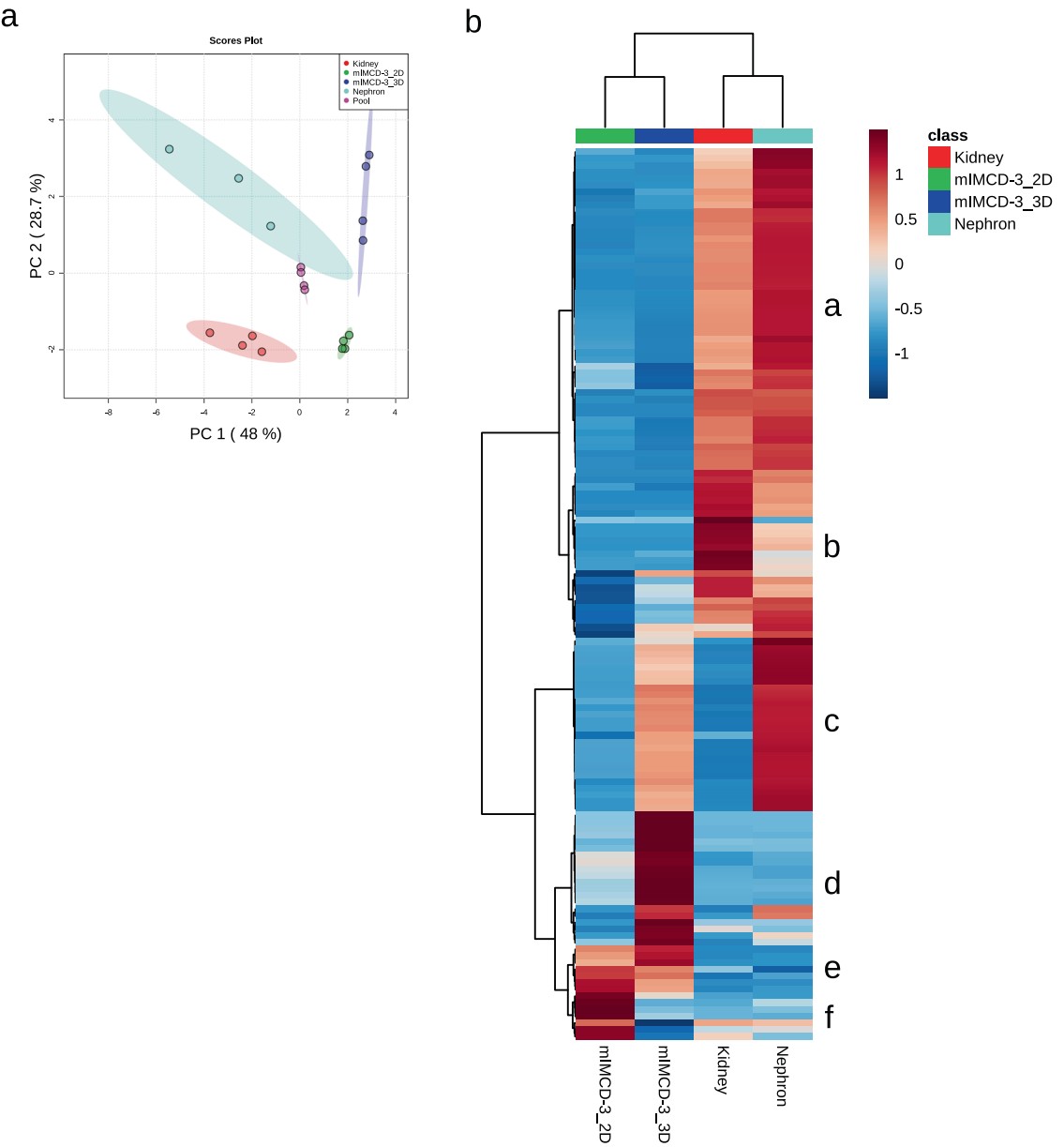

**Fig. 5 Lipid composition of cells grown in 2D of 3D or in situ. a** PCA of lipid profiling with PC1 against PC2. Red: whole kidney lysate, green: cells grown in 2D, blue: cells grown in 3D, turquoise: isolated distal tubules from perfused kidneys, and pink: pooled quality control sample. Replicates are highlighted as individual dots, shaded area shows the 95% confidence interval. **b** Heat map and cluster analysis of lipids and fatty acids. Range-scaled z-scores of annotated features with a q-value < 0.05 according to ANOVA and FDR correction. Differently abundant metabolite clusters after Pearson and Ward reflect the differences between cultured 2D or 3D grown mIMCD3 cells and kidney lysates. Green: 2D grown cells, red: whole kidney lysate, blue: 3D grown mIMCD3 cells, turquoise: isolated nephrons, n for nephrons = 3, n for 2D, 3D, and kidneys = 4. Identified lipids can be clustered into six subgroups (**a–f**, identity of the clusters is explained in the main text and Supplementary Fig. 3). A complete list of all identified entities is shown in Supplementary Data 5.

mIMCD3 cells grown in spheroids for 48 h behave similarly to in vivo cells in terms of metabolism.

Previous studies revealed that cells in 3D cultures exit active cell cycle and become dormant[11]. This change in cell cycle activities leads to a change in nutrient demand. Typically, glycolysis, the pentose-phosphate pathway, and nucleotide synthesis are upregulated when cells enter the S-phase and their activities decrease again throughout the G2-, M-, and G1-phase[33]. The metabolic pattern observed in 3D grown mIMCD3 cells was strikingly similar to cells isolated from nephrons; intermediates of the tricarboxylic acid cycle and glycolysis were strongly downregulated. This may be caused by the block in proliferation and the hypoxic environment inside the cell mass that causes the shift from aerobic glycolysis and oxidative phosphorylation. The production of macromolecules is one major function of aerobic glycolysis. To generate purine nucleotides, 10 carbon atoms are needed. They are, in part provided by 5-phosphoribosyl-α-pyrophosphate, an activated form of ribose-5-phosphate. Dormant cells require only a few new building blocks and subsequently reduce glycolysis[34].

Next, we noted a strong decrease in the pentose-phosphate-pathway in 3D grown cells and its primary murine cell counterparts. This was further confirmed by the downregulation of glucose-6-phosphate dehydrogenase on mRNA level in these

cells. The pentose-phosphate-pathway is an important source for nicotinamide adenine dinucleotide phosphate that may be used in further reductive pathways to produce fatty acids, cholesterol, nucleotides, and five carbon sugars[35].

It has to be noted that the changes in glucose uptake and processing from 2D to 3D growth are highly cell line-specific and cannot be generalized[36]. In mIMCD3 cells, the glucose levels in cell culture medium were not changed. Furthermore, it is also important to consider the effect of the missing p53 signaling in the 2D and 3D grown cells, since the mIMCD3 cells were immortalized using SV40. The tumor suppressor p53 itself is known to downregulate glycolysis, pentose phosphate pathway, liponeogenesis, and to promote oxidative phosphorylation[33]. However, we speculate that the effect of the contact inhibition on the metabolism outruns the effect of missing p53 signaling in 3D grown cells.

Besides general changes in metabolism which were most probably linked to the exit from active cell cycle[33], we also found metabolites in 3D grown cells that are characteristic for nephrons, the origin of the cells. Betaine, for instance, is an organic osmolyte that in contrast to electrolytes, does not impair protein function in high concentrations, similar to sorbitol, inositol, taurine, and glycerophophatidyl-choline[30]. Betaine is either produced by oxidation of choline or absorbed by the diet[37] and plays an important role for protein stability in hyperosmolar environments such as the kidney. Betaine and taurine have different concentrations and localizations in the kidney. While the overall kidney concentration of taurine is 1.5–20 times higher than the level of betaine, the local concentration of betaine is elevated in proximal and distal tubules and the collecting ducts[30]. This trend could also be observed in our measurements. While in 2D cell cultures only low levels of betaine and taurine were measured, the levels of betaine were raised in spheroids and in distal tubules isolated from kidneys. In the total kidney lysate though, the level of betaine was not elevated, but the level of taurine was strongly elevated, as reported elsewhere[30]. Bgt1 is a transporter for betaine. Bgt1 itself is unevenly distributed and most prominently expressed in collecting ducts but not in proximal and distal tubules[37]. The RNA and protein levels of bgt1 in kidney lysates and 3D grown cells were markedly reduced in comparison to the 2D grown cells (Fig. 4a, b). As shown in the Western blot in Fig. 4a, we saw a clear downregulation of Bgt1 in isolated nephrons that further confirms the efficiency of sample preparation.

Re-extraction of lysates with chloroform allowed for the measurement of lipids and fatty acids in a separate run. All isolated lipids clustered into six distinct clusters. In clusters a and b, vast differences appeared between in vitro grown cells and kidney derived primary cells. Interestingly, however, cluster c contains lipids and fatty acids that mostly belong to the renal vitamin D metabolism. Those were more abundant in 3D cells and nephron lining cells. In the kidney, the precursor of vitamin D, 25-hydroxyvitamin D3, is absorbed at the glomerulus, proximal and distal tubules. There, an additional hydroxyl group is added to result in 1,25-dihydroxyvitamin D3, that now may act as a transcription factor at the vitamin D receptor[38]. We also detected the catabolic intermediate 1,24-25-trihydroxyvitamin D in our screen behaving identical in 3D grown cells and nephrons suggesting a likewise regulation of CYP24A1, responsible for various catabolic reactions in vitamin D metabolism[39].

In conclusion, we show in our study that the metabolism of cells adapts to changes in the growth environment. Cells growing in 2D undergo an exponential growth regimen that completely changes when cells are contact inhibited in a 3D environment. The metabolic fingerprint of 3D grown cells is in many instances comparable to cells directly isolated from in situ isolated cells. This finding might be important for future efforts to reduce animal testing, since 3D grown cells reflect many features of in vivo grown tissues.

## Methods

**Cell culture**. Mouse inner medullary collecting duct (mIMCD3) cells were generated as reported elsewhere[40]. Cells were kept in standard 2D cell culture conditions in DMEM media, supplemented with 10% fetal calf serum and 1% penicillin/streptomycin and passaged every 3 days to keep them in a permanent exponential growth rate and avoid differentiation. To generate spheroids, cells were trypsinized, washed, counted, and diluted to a concentration of $1 \times 10^6$ cells/ml in DMEM medium. 30 µl droplets were mounted to the lid of a 15 cm cell culture dish, the lid was carefully inverted and put on the bottom of the plate that had been filled with 10 ml phosphate-buffered saline (PBS) to avoid evaporation[26,27]. 48 h after seeding, spheroids were collected by flushing the droplets with a 1 ml pipette from the lid using PBS and collected in a 50 ml reaction tube. Spheroids sank rapidly. Supernatant was carefully aspirated and spheroids were washed 5 times in PBS. Gliotoxin (AppliChem, Darmstadt, Germany) was applied at a concentration of 1 µM ultimately after seeding the spheroids.

**Kidney and nephron isolation**. All animal experiments were performed in accordance with German legislation and approved by regional authorities. For whole kidney isolation, 6–8 weeks old C57BL/6N female mice were sacrificed, kidneys were immediately explanted, and mechanically and chemically lysed in radio-immunoprecipitation assay (RIPA) buffer on ice. To isolate distal tubules, a collagenase II digestion of the kidney was performed and subsequently the distal tubules were manually sorted using a Zeiss Binocular (6× magnification, Oberkochen, Germany) equipped with a Rottermann contrast (Leica, Wetzlar, Germany)[41]. Up to 300 distal tubules were collected on ice for further analysis.

**Western blot**. To prepare spheroids for Western blot analysis, about 160 spheroids were pooled and lysed in 50 µl RIPA buffer, protein concentration was measured by Bradford assay and proteins separated by sodium dodecyl sulfate-polyacrylamide gel electrophoresis followed by transfer onto nitrocellulose membrane and Western blot analysis[25]. The antibodies used were Hexokinase 2 (Cell Signaling (C64G5) #2867S), Bgt1 (Abcam, ab200676), PCYT2 (Abcam, ab15053), Tubulin (Bio-Rad, MCA77G), GAPDH (Cell Signaling #5174), and Actin (MP Biochemicals, #69100). Secondary peroxidase-labeled anti-mouse IgG (H+L) and anti-rabbit IgG (H+L) were obtained from Jackson Immuno Research, West Grove, PA. All primary antibody dilutions were according to manufacturers' protocols, usually 1:1000. Secondary antibodies were diluted 1:5000.

**Real-time RT-PCR**. To isolate RNA from spheroids, about 80 spheroids were lysed in 400 µl of TRIzol reagent, RNA was then isolated using chloroform and isopropanol. cDNA was produced from 2 µg isolated RNA using the Super Script IV Reverse Transcriptase Kit (Thermofisher, Waltham, MA, USA)[42]. cDNA was used to run a qPCR using primers for *hexokinase2*, *bgt1*, *g6pd*, *mpcyt2α*, and *actin* (see detailed sequences in Supplementary Data 10) using MESA Blue (Eurogentec, Cologne, Germany). On a Bio-Rad CFX96 Real-Time Detection System (Bio-Rad Laboratories Inc., Hercules, CA, USA). Data is expressed as $2^{-\Delta\Delta ct}$ relative to *actin* expression and normalized to the 2D condition. For statistical analysis, one-way ANOVA with a subsequent Bonferroni's Multiple Comparisons test was engaged.

**Cell cycle analysis**. Analysis of spheroids by flow cytometry was established by washing spheroids once with PBS, then incubating spheroids for 5 min in prewarmed Trypsin-EDTA in a water bath, gently resuspending with a 1000 µl tip occasionally. After centrifugation, cells were fixed in 70% EtOH overnight at −20 °C[43]. Cells were permeabilized using 0.1% Triton X-100 and stained with a combination of 7AAD (1:1000) and anti-phospho-H3Ser10 (Cell Signaling, 1:50). Cells were analyzed using a BD LSRII (Becton Dickinson) flow cytometer and data was analyzed using FlowJo 7.6.5.

**Imaging**. Images of spheroids were captured using a ZEISS Axiovert 40C inverted Microscope (Zeiss, Jena, Germany) operated using ZEN blue imaging software and equipped with a self-designed tray to conveniently place the spheroid above the objective[27].

**Harvest for GC/MS analysis**. For two-dimensionally grown cell culture, medium was aspired and cells were washed twice with 10 ml 0.9% NaCl. Afterwards, cells were quenched by 1.5 ml ice-cold MeOH:$H_2O$ 1:1 containing 1 µg/ml ribitol and phenylglucose as internal standards and harvested by using cell scrapers. The cell suspensions were transferred into screw-cap vials prefilled with 300 mg glass beads and directly snap-frozen in liquid nitrogen.

For three-dimensionally grown cell culture, 160 droplets containing spheroids were rinsed into a test-vial by 2 ml 0.9% NaCl. Spheroids were allowed to sediment by gravity for 1 min. The medium/NaCl solution was almost completely aspirated. In this manner, spheroids were washed 5 additional times with 5 ml 0.9% NaCl.

Finally, the spheroid pellet was resuspended in ice-cold MeOH:H$_2$O 1:1 containing internal standards and transferred into screw-cap vials as described above.

Nephrons were isolated as explained beforehand[41]. Nephrons were washed twice with 0.9% NaCl, transferred together with ice-cold MeOH:H$_2$O 1:1 containing internal standards into screw-cap vials and frozen.

Kidneys were rinsed with 0.9% NaCl and weighed. 500 μl ice-cold MeOH:H$_2$O 1:1 containing internal standards were added immediately with the kidney to screw-cap vials and snap-frozen. Before tissue lysis, extraction volume was adjusted to ensure an equal tissue mass to extraction medium ratio.

**Lysis and extraction**. The cells, spheroids, and tissues were lysed by a PreCellys Evolution with 10,000 rpm for 50 s with two repetitions. Lysates were centrifuged (20,000g, 10 min, 4 °C) and metabolite-containing supernatants were transferred into a new Eppendorf tube. 500 μl CHCl$_3$ was added into the remaining vial and three rounds of 1 min vortexing at maximum intensity extracted lipids. After centrifugation (5000g, 5 min, 4 °C) lipid-containing CHCl$_3$-phase was transferred into a new Eppendorf tube. The screw-cap vials were cleared of remaining solvent in a vacuum concentrator. Afterwards, 300 μl H$_2$O for cell culture samples as well as nephrons and 900 μl H$_2$O for kidneys were added and shaken (1200 rpm, 90 min, 30 °C). DNA concentration was measured by a NanoDrop 1000 to get a rough estimation of biological material. Metabolite extracts were transferred into new vials with a volume equal to the corresponding amount of DNA across all samples. The difference in volume was balanced by addition of extraction medium before evaporation to dryness by a vacuum concentrator. Pellets were kept under nitrogen atmosphere at −80 °C until analysis.

**Untargeted GC/MS profiling**. Polar metabolites were measured as previously reported[44,45]. In brief, metabolite pellets were derivatized by methoxyamination and silylation before splitless injection to an Agilent GC/MS (7890A/5975C) equipped with an HP-5MS column (60 m). Raw spectra were processed by deconvolution and alignment[46] and annotated by comparison to different mass spectral libraries and retention index matching. Intensities were normalized to internal standard and the sum of all annotated metabolites in a sample. MetaboAnalyst 4.0 was used for statistical analysis, principal component analysis, and heat map generation[47].

Analysis of cell culture media: Metabolites from cell culture medium were extracted as previously reported by acetonitril:methanol[44]. Pellets were analyzed by GC/MS profiling in the same manner as endometabolites[48]. Instead of peak sum normalization, only the internal standard as well as a correction (17:12) for dilution introduced by the NaCl rinsing in 3D cultures was applied for normalization.

**Lipid profiling**. Lipids were analyzed with a Waters Acquity i-Class coupled to a Waters Synapt G2Si adapted from Isaac et al.[49]. Pellets were dissolved in 100 μl isopropanol:acetonitrile:water 2:1:1 and kept at 12 °C. 2 μl were injected onto a Waters BEH C18 (2.1 mm × 100 mm, 1.7 μm) column run with the following program: 60% A (A: 10 mM NH$_4$CHOO, 0.1% formic acid, 60% acetonitrile, 40% water) hold for 2 min, to 35% A until 6.5 min, to 12% A until 12 min, to 1% A until 13 min, hold for 15 min, to 60% A until 15.5 min and hold to a final time of 20 min. B was 10 mM NH$_4$CHOO, 0.1% formic acid, 10% acetonitrile, and 90% iso-propanol. The column temperature was set to 55 °C with a flow rate of 300 μl/min. The source was operated at −1.5 kV, 150 °C, sampling cone and source offset were set to 30 and 80, respectively. Desolvation temperature was 500 °C with 600 l/h desolvation gas flow and 50 l/h cone gas flow as well as 6.5 bar nebulizer pressure. For data analysis, ProgenesisQI was used and searched against HMDB database 3.6 with a minimal score of 45 which corresponds relatively to the cut-off used in the GC/MS profiling following the same statistical analysis.

**Statistics and reproducibility**. All statistical analyses regarding metabolites or lipids were conducted with MetaboAnalyst 4.0[47]. For comparison of 2D and 3D cell culture-derived endometabolites or exometabolites, an unpaired, two-tailed $t$-test was conducted, followed by multiple testing correction based on Benjamini–Hochberg. For comparison of 2D and 3D cell culture as well as nephron and kidney derived endometabolites and lipids, a one-way ANOVA was conducted, followed by multiple testing correction based on Benjamini–Hochberg. For principal component analyses and the generation of heat maps, values were range-scaled. Within the analyses of metabolites or lipids, no technical replicates were acquired, but individual samples were analyzed (four independent replicates for 2D cell culture and 3D cell culture, kidneys obtained from four mice and nephrons obtained from three mice). For monitoring technical performance and possible analytical drifts, quality control samples, which consisted of an equal mixture of all samples, were regularly injected from the same pool as technical replicates.

**Reporting summary**. Further information on research design is available in the Nature Research Reporting Summary linked to this article.

## Data availability

The datasets with all replicates of the GC/MS and LC/MS measurements and statistical analyses for endometabolite, exometabolite, and lipid analyses of this study are shown in

Supplementary Data 2–5. Raw data of flow cytometry and qPCR analysis can be provided from the corresponding author upon reasonable request, numerical quantification is listed in Supplementary Data 1.

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

## Acknowledgements

The authors thank Michael Köttgen, Uniklinik Freiburg for providing the mIMCD3 cells. C.B., L.P. and M.S. are funded by the DFG SFB 1140. S.L. and M.S. are members of the Spemann Graduate School of Biology and Medicine (SGBM) in the frame of the Excellence Initiative of the DFG. S.N. and C.B. are supported by the Research Group FOR2036 funded by the German Research Foundation (DFG). Animal experiments were performed according to the RRR principle and regulated by the Tierversuchsantrag G16/ 021 by the Regierungspräsidium Freiburg. The project was partly supported by a Bayer Foundation StartUP Grant to L.P. The authors thank the team of the animal facility CEMT for animal care. The article processing charge was funded by the University of Freiburg in the funding program Open Access Publishing.

## Author contributions

L.P. conceived the project, designed the experiments, wrote the manuscript, and edited the figures. M.S. and S.L. performed MS and GC analyses, did the statistical analysis, and drew Figs. 2, 3, and 5. A.W. and S.N. carried out the qPCR analysis. B.K. and C.B. supported the project with lab equipment and insightful comments. All authors read and approved the final manuscript.

## Competing interests

The authors declare no competing interests.
