## [Peer Review File · Communications Biology]

Reviewers' comments:

Reviewer #1 (Remarks to the Author):

The manuscript provides one more proof that cells grown as 3D cultures mimic in vivo conditions better which is not a property of the cells grown as 2D cultures. Although several earlier reports have proved that fact beyond doubt, the approach by the authors in this manuscript is unique. The uniqueness stems from the fact that they have compared 2D and 3D cultured cells with endpoints as obtained from fresh murine tissue. Till date, most reports compare 2D and 3D cultures for a variety of parameters, but, comparing these parameters or endpoints with those obtained from fresh tissue samples take the whole reasoning to a higher level.

The authors have used metabolic by products as end-points and have presented their results and discussion well. Also, the methodology is quite straight-forward and unambiguous.

The study presented in the manuscript will be of interest to others in the field, especially as a good information source to replace animal models with advanced in vitro techniques.

Reviewer #2 (Remarks to the Author):

The authors showed the adaptation of the metabolism of cells upon changes in the environment. They compared 2D cell growth, 3D spheroids and cells isolated from murine organs. The metabolome of 3D cell growth exhibited similarities to the in vivo situation. The study could provide support for the replacement of animal testing with 3D culture. It is a nice, well-organized paper that the reviewer would suggest acceptance with minor grammatical revisions.

Minor changes needed:

1. Line 24, Page 4, "The effect of e.g. a new drug" should be "The effect of, e.g., a new drug" or "The effect of a new drug, for example,"
2. Line 3, Page 16, "react entirely different" should be "react entirely differently".
3. Line 12, Page 16, "that, " should be "that".

Reviewer #3 (Remarks to the Author):

General:

The phenotypic differences observed are accompanied by differential metabolic responses between 2D cell and 3D spheroid cultures which may impact future testing and may replace animal testing. This a well-designed study, well-written, with established metabolic techniques, lacking are tracing experiments which would aid the claiming hypothesis that 3D cultures resemble the in-vivo metabolome.

Further evidence or explanation:

Cell spheroids are known to rapidly import glucose and convert it to glycogen. Could it be that when spheroids are cultured in physiological amounts of glucose (5.5 mM), the media glucose is typically exhausted (typically round 8 h) after media exchange? Thereafter, the spheroids experience 'glucose starvation' which may be closer to those present in tissues rather than in classical 2D cultures?

The metabolic reprogramming in cell spheroids may suggest an increased reliance on glutamine? Could the citrate be used for fatty acid synthesis? Could enzyme abundance of the pathways involved be explored and sustain fatty acid synthesis?

Please include statistical value in text when claiming significance.

Methods: C57BL/6, females, 6-8 weeks old, why only females? Explain. Include substrain: C57BL/6J or C57BL/6N?

Other epithelial cells behave the same, is it possible to validate the results in other cells line of epithelial origin (possible without SV40) as the resulted could be biased by SV40.

Pg.11 Line 22: Legend: Fig1B. 2D grown cell and 3D cells change their cell cycle over time. Is there any relationship with metabolism, metabolites? Is there also an exponential usage of any metabolites over time?

Pg.11 Line 24: "same amount of cells growing in 2D induced a change of medium color". Do you mean increased lactic acid? Clarification needed. As this is the basis for using 2D Vs 3D spheroids, it's quite important to clarify this statement, at this point, before other results are justified.

Pg.13 Line 21: 'Levels of hexokinase 2 were diminished on protein Fig4A'. By reference to Fig4A Tubulin (loading control) increased expression of kidney and nephron compared to 2D and 3D can be misleading to the reader. With reference to the actin protein level, there are similar levels for 2D and 3D and undetectable levels for nephron and kidney. Needs clarification and justification for which loading controls are to be used with associated set of samples. Legend to Fig 4: GADPH levels, can also be misleading, as stated in legend " is a member of glycolysis' increased protein levels are observed for GADPH in tot. kidney and 3D. Any explanation on this? This implies increased glycolysis?

Pg.13 Line 21: For Fig4C legend on text not correct, typo and refers to Fig4b, please correct. Are the mRNA levels statistically significant as in text written as "diminished". Why 2D mRNA levels all set 1? This assumes that the 2D is your control? Please clarify and include stats on graphs and text.

Fig4.Legend typo "glucose-6-phosphat dehydrogenase" add "e" to phosphate.

In text, pg14, line 6, refer to betaine would be helpful to include reference to Fig3b. Can betaine increased transport be associated when extracellular osmolality rises? This is raised in the discussion but not clear the association.

Pg.14 Line 12, stats missing for bgt-1 mRNA.

Pg.14 Line 18. PCYT2 is a gene at the junction of phospholipid and neutral lipid metabolism, can other metabolites related to lipids be informative? As supplementary Table S7, PE were detected. Please explain.

Pg.15 Line7, clusters explained in text, but would be helpful to the reader to also label the clusters in the histogram.

In discussion:

Pg18, line 11, 'clear downregulation of Bgt1'- which band refers to the downregulation of Bgt1 in the nephron? Please indicate specific band as various isoforms are observed for nephron and kidney.

Pg18 line 23, claiming regulation of CYP24A1, no reference or data is associated to this. Please explain or provide further evidence.

RESPONSE TO THE REFEREES' COMMENTS for LAGIES *et al.*

The reviewers have provided good suggestions to improve the quality of our manuscript. The manuscript has now been revised and we have addressed all of the reviewers' comments and suggestions. In the following, we detail our point-by-point responses to their comments and suggestions, for better readability we highlighted our responses in *italic*:

Reviewer #2 (Remarks to the Author):

Minor changes needed:

1. Line 24, Page 4, "The effect of e.g. a new drug" should be "The effect of, e.g., a new drug" or "The effect of a new drug, for example,"

Thank you, we fixed this grammar mistake.

2. Line 3, Page 16, "react entirely different" should be "react entirely differently".

Thank you, we fixed this grammar mistake.

3. Line 12, Page 16, "that, " should be "that".

Thank you, we fixed this punctuation mistake.

Reviewer #3 (Remarks to the Author):

Further evidence or explanation:

Cell spheroids are known to rapidly import glucose and convert it to glycogen. Could it be that when spheroids are cultured in physiological amounts of glucose (5.5 mM), the media glucose is typically exhausted (typically round 8 h) after media exchange? Thereafter, the spheroids experience 'glucose starvation' which may be closer to those present in tissues rather than in classical 2D cultures?

We thank the reviewer for this suggestion. We agree that a physiological concentration of glucose in the media mimics the physiological situation even better. However, we cultured the mIMCD-3 cells in normal DMEM, containing 25 mM glucose as this is the most common condition researchers grow mIMCD-3 cells in. In addition, a rapid conversion of glucose to glycogen is a predominant process in liver cells (1). Following your remark about medium color change, we analyzed metabolites in cell culture medium (kindly see below). We could not detect a difference in glucose levels between 2D and 3D conditioned media (q-value: 0.6409). Therefore, we do not think that a starvation led to the observed closer resemblance of 3D grown renal cells to nephrons.

The metabolic reprogramming in cell spheroids may suggest an increased reliance

on glutamine? Could the citrate be used for fatty acid synthesis? Could enzyme abundance of the pathways involved be explored and sustain fatty acid synthesis?

Several studies have already aimed at investigating the metabolic differences induced by 3D cell culture techniques and proposed a reductive glutamine metabolism to fuel citric acid/acetyl-CoA (2) . Therefore, we agree with the reviewer that glutamine and citric acid is probably also used for fatty acid synthesis in mIMCD-3 cell spheroids but cannot give any further evidence for this.

Please include statistical value in text when claiming significance.

We have now provided the corresponding statistical value whenever appropriate.

Methods: C57BL/6, females, 6-8 weeks old, why only females? Explain. Include substrain: C57BL/6J or C57BL/6N?

From our experience we know that the urine of male mice is much more laden with excretory metabolites such as pheromones. We were worried that these compounds will confound our metabolome analysis. Additionally, we wanted to keep the total number of mice low in respect to the 3R-principle. A mixed population would have risen the number to account for inter-sex related variations. That is why we only concentrated on the analysis of female kidneys.

We used the C57BL/6N substrain.

Other epithelial cells behave the same, is it possible to validate the results in other cells line of epithelial origin (possible without SV40) as the resulted could be biased by SV40.

This study focuses on the metabolism in the kidney. The initial idea was to validate, if established, widely in the field used cell lines still show any similarity to freshly isolated cells from the same area, where the cell lines were initially isolated. In nephrology, mIMCD3, MDCK and HEK293T cells are the standard workhorses, where only the mIMCD3 cell line is of murine origin and therefore was chosen to compare to murine tissue. We are aware that the immortalization using SV40 bears a lot of dangers, however we tried several other methods to yield a stable cell line of epithelial cells from the kidney but all of them failed. Alternative immortalization methods such as spontaneous immortalization or the 3T3 approach did not result into any viable cell lines. Furthermore, I am personally not a big fan of spontaneous immortalization, since the driving mutation behind the immortality is unknown to the researcher (usually they rest in the p53, RAS or RB pathways). With the SV40 virus one at least knows, that all p53 regulated pathways needs to be handled with care.

Pg.11 Line 22: Legend: Fig1B. 2D grown cell and 3D cells change their cell cycle over time. Is there any relationship with metabolism, metabolites? Is there also an exponential usage of any metabolites over time?

There is decidedly a cross-talk between metabolism and different cell cycle phases. We had already indicated this in the discussion. We now extended the description of cell cycle accompanying metabolic changes in the same paragraph, which are in line with our findings. To provide sufficient energy and building blocks for DNA synthesis, glucose is especially needed during the S-phase. We detected elevated intermediates of glycolysis in the 2D grown cells as well as elevated mRNA levels of involved enzymes, which proves elevated activity in these pathways (glycolysis and PPP). However, extracellular glucose was not changed (q-value: 0.6409; please also review our response to glucose and glycogen above).

Pg.11 Line 24: “same amount of cells growing in 2D induced a change of medium color”. Do you mean increased lactic acid? Clarification needed. As this is the basis for using 2D Vs 3D spheroids, it’s quite important to clarify this statement, at this point, before other results are justified.

We thank the reviewer for this remark. Indeed, the acidification could be caused by many molecules, of which lactic acid is of course a highly probable one. Therefore, we started to investigate the metabolic changes going along with the different growing conditions. We took your remark about the importance of this observation to include metabolomics data of cell culture medium. The lactic acid levels were indeed significantly increased (q-value: 0.0002), which confirms our findings about the intracellular metabolites.

Pg.13 Line 21: ‘Levels of hexokinase 2 were diminished on protein Fig4A’. By reference to Fig4A Tubulin (loading control) increased expression of kidney and nephron compared to 2D and 3D can be misleading to the reader. With reference to the actin protein level, there are similar levels for 2D and 3D and undetectable levels for nephron and kidney. Needs clarification and justification for which loading controls are to be used with associated set of samples. Legend to Fig 4: GAPDH levels, can also be misleading, as stated in legend “is a member of glycolysis’ increased protein levels are observed for GAPDH in tot. kidney and 3D. Any explanation on this? This implies increased glycolysis?

We are truly aware, that the Westernblot in Figure 4A does not meet the expectation of the trained eye of a researcher. However, we struggled a lot to produce the perfect blot, with all lanes showing comparable lines in the loading control. However, it seems, that cells growing in cell culture for many generations change their actin, gapdh and tubulin content in comparison to freshly isolated cells. We put special effort in yielding a satisfactory result for the Bgl1 signal, since its regulation could have been nicely showed in the MS and qPCR data.

Figure 1: Westernblot on lysates from either 2D or 3D grown cell culture cells (mIMCD3) or freshly isolated cells from isolated nephrons or entire kidney lysates. Arrow indicates calculated size of Bgt1.

However, the Westernblot shown in Figure 1 still does not show even loading in the housekeeper proteins.

Increased protein levels of GAPDH indeed also suggests increased glycolysis. However, decreased hexokinase levels as well as decreased intracellular key metabolites in glycolysis and the reduced lactate levels in the medium let us to the conclusion of reduced glycolysis in 3D grown cells. Although GAPDH is best known for its role in glycolysis, it has several other functions within the cell (3). However, we cannot explain which factors led to this contrary expression.

Pg.13 Line 21: For Fig4C legend on text not correct, typo and refers to Fig4b, please correct. Are the mRNA levels statistically significant as in text written as “diminished”. Why 2D mRNA levels all set 1? This assumes that the 2D is your control? Please clarify and include stats on graphs and text.

Thank you for pointing out this mishap. We fixed the typo and the confusion in figure order.

Since we used the $2^{-\Delta\Delta Ct}$ method, we set the 2D cell culture values as 1 and compared the other three samples relative to the 2D value. Technical repeats were averaged in one data point, biological replicates with measurements on different days were printed at individual dots. Of course, intra-experimental variation gets lost by doing this; we can easily provide this upon request. Statistics are now displayed.

Fig4.Legend typo “glucose-6-phosphat dehydrogenase” add “e” to phosphate.

Thank you for the hint. We fixed that

In text, pg14, line 6, refer to betaine would be helpful to include reference to Fig3b. Can betaine increased transport be associated when extracellular osmolality rises? This is raised in the discussion but not clear the association.

We thank the reviewer for this notification and added the reference to figure 3b. The reviewer is absolutely right about the regulation of the betaine transporter and osmolality per se (4). In our study however, we did not include hyperosmolar conditions in the cell culture and neither is betaine present in normal DMEM at substantial concentrations. This is the reason why we mentioned the association of betaine transport and osmolality only in the discussion but rather think that, in the case of 3D grown mIMCD-3 cells, betaine is primarily synthesized.

Pg.14 Line 12, stats missing for bgt-1 mRNA.

Stats were inserted. We performed a one-way ANOVA with a Bonferroni Post Test. The decrease in 3D and mouse tissue was highly significant ($p < 0.0001$) in comparison to 2D control cells.

Pg.14 Line 18. PCYT2 is a gene at the junction of phospholipid and neutral lipid metabolism, can other metabolites related to lipids be informative? As supplementary Table S7, PE were detected. Please explain.

GC/MS measurements revealed ethanolaminephosphate was increased in 3d grown cells compared to the 2D condition. PCYT2 encodes for the enzyme CTP:phosphoethanolamine cytidyltransferase, which catalyzes the reaction from CTP + ethanolaminephosphate to CDP-ethanolamine + pyrophosphate (5). Phosphatidylethanolamine (PE) is synthesized from CDP-ethanolamine and diacylglycerol and is one of the major content in mammalian cell membranes. PE can additionally be synthesized by decarboxylation of phosphatidylserine. Therefore, it is reasonable that PE-species were detected despite the inconclusiveness regarding the regulation on protein, transcript and metabolite level.

Pg.15 Line7, clusters explained in text, but would be helpful to the reader to also label the clusters in the histogram.

We now provide heat map data with all species labeled. However, due to the high number of lipid features, we provide the labelled clusters separately in the supplementary material to ensure readability.

In discussion: Pg18, line 11, 'clear downregulation of Bgt1' - which band refers to the downregulation of Bgt1 in the nephron? Please indicate specific band as various isoforms are observed for nephron and kidney.

Thank you for pointing out that issue. We now indicate the correct band of the Protein as suggested by the manufacturer's instruction.

Pg18 line 23, claiming regulation of CYP24A1, no reference or data is associated to this. Please explain or provide further evidence.

We inserted references highlighting the role of CYP24A1 in renal vitamin D metabolism. Thereby, we noticed that we have introduced a construing error. 24,25-Dihydroxyvitamin D was not found in primary nephrons and 3D grown mIMCD-3 cells, but 1,24,25-trihydroxyvitamin D, which are both metabolized by CYP24A1. We sincerely apologize for this lapse.

We have carefully examined the manuscript. We sincerely hope that the manuscript has been revised to your satisfaction. It would give us a great pleasure to see the acceptance of the revised manuscript for publication in *Communications Biology*.

With best regards,

Lukas Peintner

References:

1. Wrzesinski K, Fey SJ. Metabolic Reprogramming and the Recovery of Physiological Functionality in 3D Cultures in Micro-Bioreactors. *Bioengineering*. 2018;5(1).
2. Jiang L, Shestov AA, Swain P, Yang C, Parker SJ, Wang QA, et al. Reductive carboxylation supports redox homeostasis during anchorage-independent growth. *Nature*. 2016;532(7598):255-8.
3. Sirover MA. Pleiotropic effects of moonlighting glyceraldehyde-3-phosphate dehydrogenase (GAPDH) in cancer progression, invasiveness, and metastases. *Cancer and Metastasis Reviews*. 2018;37(4):665-76.
4. Kempson SA, Montrose MH. Osmotic regulation of renal betaine transport: transcription and beyond. *Pflügers Archiv*. 2004;449(3):227-34.
5. van der Veen JN, Kennelly JP, Wan S, Vance JE, Vance DE, Jacobs RL. The critical role of phosphatidylcholine and phosphatidylethanolamine metabolism in health and disease. *Biochimica et Biophysica Acta (BBA) - Biomembranes*. 2017;1859(9, Part B):1558-72.